# Long Term Assessment of Anti-SARS-CoV-2 Immunogenicity after mRNA Vaccine in Persons Living with HIV

**DOI:** 10.3390/vaccines11121739

**Published:** 2023-11-22

**Authors:** Alessandra Vergori, Alessandro Cozzi-Lepri, Giulia Matusali, Stefania Cicalini, Veronica Bordoni, Silvia Meschi, Valentina Mazzotta, Francesca Colavita, Marisa Fusto, Eleonora Cimini, Stefania Notari, Veronica D’Aquila, Simone Lanini, Daniele Lapa, Roberta Gagliardini, Davide Mariotti, Giuseppina Giannico, Enrico Girardi, Francesco Vaia, Chiara Agrati, Fabrizio Maggi, Andrea Antinori

**Affiliations:** 1HIV/AIDS Unit, National Institute for Infectious Diseases L. Spallanzani, Istituto di Ricovero e Cura a Carattere Scientifico (IRCCS), 00149 Rome, Italy; alessandra.vergori@inmi.it (A.V.); stefania.cicalini@inmi.it (S.C.); valentina.mazzotta@inmi.it (V.M.); marisa.fusto@inmi.it (M.F.); simone.lanini@inmi.it (S.L.); roberta.gagliardini@inmi.it (R.G.); giuseppina.giannico@inmi.it (G.G.); andrea.antinori@inmi.it (A.A.); 2Centre for Clinical Research, Epidemiology, Modelling and Evaluation (CREME), Institute of Global Health, University College London, London NW3 2PF, UK; 3Laboratory of Virology, National Institute for Infectious Diseases L. Spallanzani, Istituto di Ricovero e Cura a Carattere Scientifico (IRCCS), 00149 Rome, Italy; giulia.matusali@inmi.it (G.M.); silvia.meschi@inmi.it (S.M.); francesca.colavita@inmi.it (F.C.); daniele.lapa@inmi.it (D.L.); davide.mariotti@inmi.it (D.M.); fabrizio.maggi@inmi.it (F.M.); 4Unit of Pathogen Specific Immunity, Department of Paediatric Haematology and Oncology, Bambino Gesù Children’s Hospital, Istituto di Ricovero e Cura a Carattere Scientifico (IRCCS), 00149 Rome, Italy; veronica.bordoni@opbg.net (V.B.); chiara.agrati@opbg.net (C.A.); 5Laboratory of Cellular Immunology and Pharmacology, National Institute for Infectious Diseases L. Spallanzani, Istituto di Ricovero e Cura a Carattere Scientifico (IRCCS), 00149 Rome, Italy; eleonora.cimini@inmi.it (E.C.); stefania.notari@inmi.it (S.N.); 6Department of System Medicine, Faculty of Medicine, Tor Vergata University, 00133 Rome, Italy; v.daquila993@gmail.com; 7Scientific Direction, National Institute for Infectious Diseases L. Spallanzani, Istituto di Ricovero e Cura a Carattere Scientifico (IRCCS), 00149 Rome, Italy; enrico.girardi@inmi.it; 8General Directorate of Prevention, Ministry of Health, 00144 Rome, Italy; f.vaia@sanita.it

**Keywords:** HIV-1 infection, SARS-CoV-2 infection, neutralizing antibodies, mRNA vaccines, T cell immunity, immunity waning

## Abstract

(1) Background: Waning of neutralizing and cell-mediated immune response after the primary vaccine cycle (PVC) and the first booster dose (BD) is of concern, especially for PLWH with a CD4 count ≤200 cells/mm^3^. (2) Methods: Neutralizing antibodies (nAbs) titers by microneutralization assay against WD614G/Omicron BA.1 and IFNγ production by ELISA assay were measured in samples of PLWH at four time points [2 and 4 months post-PVC (T1 and T2), 2 weeks and 5 months after the BD (T3 and T4)]. Participants were stratified by CD4 count after PVC (LCD4, ≤200/mm^3^; ICD4, 201–500/mm^3^, and HCD4, >500/mm^3^). Mixed models were used to compare mean responses over T1–T4 across CD4 groups. (3) Results: 314 PLWH on ART (LCD4 = 56; ICD4 = 120; HCD4 = 138) were enrolled. At T2, levels of nAbs were significantly lower in LCD4 vs. ICD4/HCD4 (*p* = 0.04). The BD was crucial for increasing nAbs titers above 1:40 at T3 and up to T4 for WD614G. A positive T cell response after PVC was observed in all groups, regardless of CD4 (*p* = 0.31). (4) Conclusions: Waning of nAbs after PVC was more important in LCD4 group. The BD managed to re-establish higher levels of nAbs against WD614G, which were retained for 5 months, but for shorter time for Omicron BA.1. The T cellular response in the LCD4 group was lower than that seen in participants with higher CD4 count, but, importantly, it remained above detectable levels over the entire study period.

## 1. Introduction

Although the availability of an effective antiretroviral therapy (ART), immunopathology in HIV-infected persons, characterized by an increased immune activation and phenotypic exhaustion, remains significantly high compared to HIV-uninfected subjects [1,2,3,4]. This raises concerns about the ability of vaccines to elicit a robust immune response against SARS-CoV-2 and its durability, necessary to confer protection from severe COVID-19.

Vaccine-induced immunity remains highly effective in preventing severe COVID-19 [5], and this is likely to be mediated by spike-specific T cells, whose inductions have been shown to correlate with vaccine efficacy [6] and are largely unaffected by viral mutations [7,8,9,10]. To date, much more importance is given to neutralization antibodies titers (nAbs), accepted by regulatory authorities as a correlate of protection and defined as the immune marker that can be used to reliably predict the level of efficacy of a vaccine in preventing a clinical outcome [11].

However, to date, few studies evaluated the durability of immunity following the vaccination against COVID-19 in PLWH [12,13,14], and no studies have reported a comprehensive analysis of the durability of neutralizing activity and T cell responses after a primary vaccine cycle (PVC) and after a first booster dose. Studies on T cellular responses to these vaccines are technically difficult, and the data are particularly sparse. However, given the possible T cell dysfunction of HIV infection, it would be of utmost importance to assess the ability of COVID-19 vaccines to induce and maintain SARS-CoV-2–specific T cells immunity, which has been shown to give protection against the severe forms of COVID-19 [15].

PLWH with a high CD4 count, similarly to what seen in HIV-negative individuals [16,17], show waning antibody immunity but persistent T cell responses six months post-vaccination [18,19], which might be due to vaccine-elicited cellular immune memory in ensuring a long-term protection. Whether this is also true for PLWH with a CD4 count <500 cells/mm^3^ remains to be established.

Furthermore, according to non-linear dynamic models [20], humoral responses in the general population following COVID-19 vaccination decreased in all age groups after six months and a continuous waning in humoral responses was estimated primarily in the older population and in individuals with a delayed administration of the second vaccine dose. Still, in the general population, natural infection prior to completion of vaccination induced both more stable anti SARS-CoV-2 IgG and neutralizing antibodies (nAbs) titers, and higher levels of IFN-γ release from activated T cells, so prior infection is a confounding factor when examining the response to vaccine [20].

In a cohort of PLWH, and according to their CD4 count one month after PVC, we evaluated the trajectories of nAbs and IFN-γ production in response to spike stimulation over four fixed time points after PVC with focus on the rate of waning after PVC and after the first booster dose (BD). 

## 2. Materials and Methods

### 2.1. Study Design and Population

The HIV-VAC study is an observational cohort study on the outcomes of COVID-19 mRNA vaccination in PLWH conducted at the Lazzaro Spallanzani INMI hospital in Roma, Italy. Details of this study have been described elsewhere [21]. In brief, demographic, epidemiologic, clinical and laboratory characteristics of PLWH undergoing a vaccination program consisting in the initial PVC followed by a BD were collected. For this analysis, which focused on the rate of waning after PVC and BD, we only considered response markers measured at the following four time points: 1- and 4-months post-PVC (T1 and T2); 2 weeks and 5 months after the BD (T3 and T4) (Figure 1).

The study was conducted according to the guidelines of the Declaration of Helsinki and was approved by the Scientific Committee of the Italian Drug Agency (AIFA) and by the Ethical Committee of the Lazzaro Spallanzani Institute, as National Review Board for COVID-19 pandemic in Italy (approval number 423/2021; amendment adopted with no. 91/2022). 

In the present analysis, the study population consisted of PLWH with no currently ongoing opportunistic infections who: (i) mostly completed the two-dose PVC with BNT162b2 or mRNA-1273 vaccines (there were 5 participants who received an adenovirus vector vaccine); (ii) received the first booster dose (BD; third vaccine shot, all mRNA); (iii) had repeated measurements of immune-response parameters available at T2 and at ≥1 of the other 3 time points in study: 2 months post-PVC (T1), 2 weeks (T3), and 5 months after the BD (third vaccine shot) (T4). Individuals with a natural SARS-CoV-2 infection diagnosis, defined by a RT-PCR positive on the nasopharyngeal swab, or positivity to anti-N and/or to anti-S/RBD antibodies at the time of the first dose of vaccine or to anti-N at any time point, were excluded from this analysis. Participants were stratified according to the CD4-T cell count registered 1 month after PVC into three groups as follows: (1) patients with CD4-T cell count <200/mm^3^ (Low CD4 count, LCD4); (2) patients with CD4-T cell count between 200 and 500 cell/mm^3^ (intermediate CD4 count, ICD4); (3) patients with CD4-T cell count > 500/mm^3^ (high CD4 count, HCD4).

### 2.2. Laboratory Procedures

Micro-neutralization assay (MNA) was performed as previously described, using SARS-CoV-2/Human/ITA/PAVIA10734/2020 and BA.1 (GISAID accession ID EPI_ISL_7716384) as challenging virus [22]. Briefly, serum samples were heat-inactivated at 56 °C for 30 min and titrated in duplicate in 7 two-fold serial dilutions (starting dilution 1:10). Equal volumes (50 μL) of serum and medium containing 100 TCID50 SARS-CoV-2 were mixed and incubated at 37 °C for 30 min. Serum-Virus mixtures were then added to sub-confluent Vero E6 cell monolayers and incubated at 37 °C and 5% CO_2_. After 48 h, microplates were observed by light microscope for the presence of cytopathic effect (CPE). The highest serum dilution inhibiting 90% of CPE was defined as MNA90. To standardize inter-assay procedures, positive control samples showing high (1:160) and low (1:40) neutralizing activity were included in each assay session. Serum from the National Institute for Biological Standards and Control (South Mimms, UK; NIBSC) with known neutralization titer (Research reagent for anti-SARS-CoV-2 Ab NIBSC code 20/130) was used as reference in MNA. The cut-off of MNA90 > 1:10 was used to define the presence of neutralizing activity, samples resulted ≥1:640 were arbitrarily considered =1:1280. In the analyses, we defined as sub-optimal a neutralizing activity of MNA90 < 1:40, as discussed in Matusali et al. [23]. We studied IFN-γ production in response to spike stimulation as a surrogate of specific T cell function. Briefly, 1 mL of heparinized whole blood was stimulated in vitro at 37 °C (5% CO_2_) with a pool of peptides covering the sequence of the wild type SARS-CoV-2 spike protein (SARS-CoV-2 PepTivator^®^ Prot_S1, Prot_S, and Prot_S+, Miltenyi Biotec, Bergisch Gladbach, Germany). After 16–20 h of incubation, plasma was harvested and stored at −80 °C until use. IFN-γ levels were measured by an automatic ELISA (ELLA, protein simple), and the IFN-γ values obtained from the stimulated samples were subtracted from the unstimulated-control value. The Staphylococcal enterotoxin B (SEB) was used as a positive control. The detection limit of these assays was 0.17 pg/mL for IFN-γ and the cut-off used in this analysis to define the T-specific cell response was 12 pg/mL, calculated as the mean +2DS of the response to spike peptides of unvaccinated uninfected healthy donors [24]. For computational and statistical purposes, nAbs titers were expressed as the log_2_ of the reciprocal of serum dilution achieving MNA90, and IFN-γ as the log_2_ of measured pg/mL.

CD4-T cell counts were addressed by flow cytometry (Aquios, Beckman Coulter, CA, USA) [24].

### 2.3. Statistical Analyses

Characteristics of patients by CD4 count groups were described and reported as the number of participants with relative frequencies for categorical factors and as the median and interquartile range (IQR) for continuous variables. Repeated measurements of immune-response parameters were available at four fixed time points of the study (T1–T4). We used the threshold (expressed in log_2_) of 5.32 log_2_ (1:40) to define a good level of neutralization and the threshold of 3.58 log_2_ for a detectable cellular mediated response (equivalent to 12 pg/mL) [23,24]. Levels below these thresholds were considered as suboptimal responses. Box-plots were used to describe the distribution of the raw data over T1–T4. The Kruskal-Wallis test was used to compare the ranking distributions of the immune responses by time and CD4 count groups. The Spearman rho statistic was used to evaluate the correlation between nAbs and IFN-γ responses at T1 and T3, overall and after stratification by CD4 count groups. We then performed unadjusted and adjusted parametric mixed linear models with random intercept and slopes including the main effects (CD4 count group and time) as well as the interaction term. Multivariable models included the three main identified potential measured confounding factors: gender, age, and CD4 count nadir. We fitted separate models for each of the parameters and estimated at each time point the mean levels and difference by CD4 count groups with the corresponding 95% CI. Unadjusted and adjusted estimates are reported both in Tables and Figures. We conducted 3 symmetric analyses, the first using as the outcome the absolute parameters values and a second analysis using the changes from T2 as the main response. This second analysis and the related statistical tests essentially compare the slopes over T1–T2, T3–T2, and T4–T2 by CD4 count group. Each of these contrasts has a specific meaning: T1–T2 tests the rate of waning over the 4 months post-PVC; T3–T2 tests the boosting effect of the 3rd dose and finally T4–T2 tests the waning of the third dose by 5 months from BD compared to what was achieved with PVC. Finally, because we were also interested in estimating and comparing the waning after the peak response achieved with the 3rd dose (T4–T3) by CD4 groups, we fitted a third model with outcome changes from T3 instead of T2. In all mixed linear model analyses specific contrasts (i.e., the mean nAbs difference by CD4 count groups at times T1–T4) were highlighted only when there was an overall type-3 significant *p*-value (<0.05) for the global test for interaction between CD4 count exposure group and time. Analyses were repeated for both the Wuhan-D614 and Omicron BA.1 measures. All the above-mentioned analyses were performed by SAS version 9.4 (Carey North Caroline USA) Prism 6.0 (GraphPad, La Jolla, CA, USA) and STATA 13.0 (College Station, TX, USA) software.

## 3. Results

### 3.1. Study Population

A total of 314 PLWH with a measure of Wuhan-D614 (WD614G) nAbs/IFN-γ at time T2 (4 months after PVC) and ≥1 measure at one of the remaining time points were included in the analysis (LCD4 = 56; ICD4 = 120; HCD4 = 138); they were those who received the PVC and, as per study protocol, returned to receive their first BD (third vaccine shot). Not all 314 participants contributed a complete set of response values at all four time points. For example, at T2, we were able to measure Omicron BA.1 nAbs and IFN-γ only in 103 (33%) and 299 (95%) participants, respectively. Appendix A describes the number of markers used at each of the four time points overall and by CD4 count groups. Missing markers values were due to participants own decision not to continue in the study or to technical reasons. The main characteristics at T2 of the 314 HIV-infected participants according to CD4 T cell count groups are reported in Table 1. Briefly, median age was 56 years (IQR 50, 61), all participants were on ART, 95% had HIV-RNA < 50 copies/mL with a median time since HIV diagnosis of 9 years (4–21) and of 5 years (4–8) since AIDS, if diagnosed, with a median number of comorbidities of 1 (1–2). The breakdown of participants according to type of vaccine used for PVC was as follows: 156 BNT162b2 (50%), 153 mRNA1273 (49%) and the remaining 5 (1%) adenoviral vector vaccines. For the 3rd dose the vaccine used was mRNA1273 for 66% of participants and BNT162b2 for 34%. Overall, the median time from the date of PVC completion to first response evaluation (T1) was 57 days (IQR: 51–58); there was evidence this length of time was slightly longer in the CD4 count > 200/mm^3^ groups (58 days, IQR: 51–59) vs. the LCD4 group (51; IQR:51–53, Kruskal Wallis test *p* = 0.005). The median distance between subsequent time-points were 119 days (108–130; T1–T2), 17 days (14–21; T2–T3) and 146 days (143–157; T3–T4), respectively. We also found strong evidence for a difference in CD4 count nadir (*p* < 0.001) and time since HIV diagnosis (*p* = 0.007) by CD4 groups. Likewise, the proportion of PLWH with HIV-RNA ≤ 50 copies/mL at T2 was 79% in LCD4, 98% in ICD4, and 99% in HCD4 (*p* < 0.001). The distribution of response markers over T1–T4 are shown as median and IQR by means of the box-plots in Figure 2. 

### 3.2. Neutralization Activity

Concerning nAbs against WD614G, at all the time points there was a clear relationship with CD4 count, with nAbs titers in the HCD4 group being higher than those in ICD4, which, in turn, were higher than those seen for LCD4 (for example at T1: 6.32 log_2_ HCD4 vs. 5.32 ICD4 vs. 3.32 LCD4; Kruskal-Wallis *p*-value < 0.0001, Figure 2). Of note, over T1–T2, WD614G nAbs levels in LCD4 remained below 5.32 log_2_, the chosen threshold for neutralization, while for HCD4 this value was 6.32 log_2_ already at T1 (Figure 2). This was confirmed by estimates from the mixed linear model, after adjusting for age and CD4 count nadir, showing also that the BD was crucial for increasing the nAbs average levels and retaining them above suboptimal neutralization until T4 in all CD4 count groups (Figure 2). From fitting this same multivariable mixed-linear regression model, we found evidence for a significant difference in nAbs trajectories over time by CD4 count groups (interaction *p* = 0.04). When we investigated contrasts at specific time-points, using HCD4 as the comparator, LCD4 showed a significantly lower mean WD614G nAbs at all time points: T1 [−2.6 (−3.3, −1.8); *p* < 0.001], at T2 [−2.1 (−2.7, −1.5); *p* < 0.001], at T3 [−1.4 (−2.1, −0.7); *p* < 0.001] and at T4 [−1.2 (−2.0, −0.3); *p* = 0.006, Table 2A]. From the analysis of the changes from T2, there was no evidence for a difference between groups in the changes of nAbs over T1–T2 (Table 2B). In contrast, the increase over T2–T3 was 0.9 log_2_ larger in HCD4 as compared to ICD4/LCD4 (*p* < 0.002, Table 2B). Similarly, at T4, nAbs remained above the levels observed at T2 in all CD4 groups, but the difference was 0.8 log_2_ larger for HCD4 vs. LCD4/ICD4 (*p* < 0.02, Table 2B). Finally, nAbs appeared to decrease by 0.2 log_2_ faster in LCD4 vs. HCD4 over T3–T4, but the difference was not significant (*p* = 0.68) (Table 2C). Of note, concerning Omicron BA.1 nAbs (Appendix A, response levels were undetectable (<3.32 log_2_) over T1–T2 in all CD4 count groups. The third dose again was pivotal for increasing these levels above suboptimal neutralization at T3, but waning over T3–T4 reset the levels to below 5.32 log_2_ at T4 in all groups (Appendix A. For this outcome, there was little evidence for a difference in trend by group over time (interaction *p* = 0.13, Appendix A).

### 3.3. IFN-γ

At all the time points identified, IFN-γ levels were above the threshold of 3.32 log_2_ (12 pg/mL) for the duration of the study and regardless of CD4 count groups (Figure 3. Mean values of spike-specific T cell response in HCD4 and ICD4 were higher than those seen for LCD4 (for example at T3: 8.2 log_2_ HCD4 vs. 7.7 (*p* = 0.09) ICD4 vs. 5.3 LCD4 *p* < 0.001; Table 3A, Figure 3) suggesting a stronger T cell function in HCD4. Of note, no evidence was observed from the mixed-linear model for a difference in time trajectories by CD4 count group (interaction *p*-value = 0.31, Table 3A,B).

### 3.4. Correlation between WD614G nAbs and IFN-γ Response Both at Time T1 and T3, Overall and after Stratification for CD4 Count Groups

We found a significant correlation between humoral and T cell responses (*p* < 0.0001), especially in the cross-sectional analyses correlating values measured at the same time which was stronger at T1 (Spearman rho = 0.48) vs. T3 (rho = 0.37). Consistently with the main results, the plot, after stratification for CD4 count groups, showed lower responses in both nAbs and IFN-γ were seen in the LCD4 vs. the HCD4 group, also leading to stronger correlations with steeper slopes (Appendix A).

## 4. Discussion

Our data showed PLWH with low CD4 count at the end of PVC had lower values of nAbs vs. ICD4 and HCD4 at all timepoints during the study and that there was a CD4-response relationship with nAbs waning over time (faster decline in the LCD4 vs. HCD4 group). The additional BD appeared pivotal for increasing the average of these levels for all CD4 count groups, also for the subvariant omicron BA.1. Advanced PLWH (low CD4+T cell counts, detectable viremia, and/or previous AIDS) have weaker humoral responses to mRNA vaccines [21,25,26,27,28,29,30,31], suggesting they might benefit from additional vaccine doses in order to increase plasma antibody concentrations and subsequently achieve an optimal nAbs titer that could be effective against SARS-CoV-2 infection. This seems particularly important if the vaccine used is not specific to elicit responses against new circulating variants of concern (VoCs) as the waning rate against BA.1 was quicker. Indeed, although estimates lack precision, it is well known vaccine efficacy wanes after PVC and can be enhanced by a booster vaccine dose and higher antibody titers are associated with higher efficacy [32]. Waning immunity, also after a booster dose, has been associated with increased vulnerability to SARS-CoV-2 infection/reinfection, particularly in case of immune-evasive VOCs [33,34,35]. Our results are consistent with other reports that showed a similar decay of humoral immune responses after six months in PLWH with high CD4 count and in healthy donors following SARS-CoV-2 vaccinations [18,19,27,36], but the peculiarity of this work is the inclusion of participants with CD4 count <500 cells/mm^3^. Of note, responses in this latter group appear to be similar to those seen in the general population. Indeed, when humoral response was evaluated by the same methods in another analysis conducted at our Institute in a cohort of healthy individuals without HIV infection (health care workers -HCW) a significant reduction of anti-RBD titer was observed after three months with a similar waning kinetic for neutralizing antibody [17]. Similarly concerning the titers post-third dose, we compared our results with those of a study conducted at our institute after performing the same tests and Wuhan nAbs titers in HCD4 two-weeks after the third dose (8.8 log_2_), these were comparable to those of healthy individuals (HCW) 8.3 log_2_ by one month post-third dose [37].

In addition, we also evaluated the levels of neutralizing antibodies over time and according to CD4 count at the end of PVC, which is the marker currently accepted as a primary correlate of protection against COVID-19. Nonetheless, memory B- and T cells—which modulate adaptive immune responses, serve as a tool of defense against disease severity and may exhibit greater durability [38] and this is why a lot of attention has been given to cell-mediated immune responses post-COVID-19 vaccination. Although T cell responses were previously evaluated only in small studies, persistent spike-specific T cells six-months post-primary mRNA vaccination were found among older PLWH [36], a response which retained activity against viral mutations [9]. In our study, IFN-γ production, which was used as a surrogate of T specific cell immunity, was lower in the LCD4 compared to the other groups but remained stable over time and above the threshold of suboptimal response regardless of CD4 count. These data confirm the importance of PVC in the induction of a protective effect from severe COVID-19, regardless of current CD4 count, despite the observed dramatic decline of humoral responses, which were on average below the optimal threshold already by four months from the second dose. These findings are also consistent with other previous data showing vaccination protects against hospitalization even up to six months after the injection [16,39]. Thus, despite an increased risk of getting infected because of the waned humoral response, the risk of severe disease in case of infection with SAR-CoV2 remains low. 

Similarly, to the humoral response, this finding is in line with an analysis conducted at our institute in a cohort of healthy individuals without HIV infection (health care workers -HCW) assessing T cells response by the same method: differently from the antibody titer, specific T cells persisted over time [17]. For what concerns the T cells immunity after the third dose, no differences in a similar study conducted in HCW were found: 8.2 log_2_ in HCDR vs. 8.3 in HCWs [27].

Our analysis has some limitations to be acknowledged. First, this is an observational setting and for the association analysis with CD4 count, unmeasured confounding bias cannot be ruled out. Second, the study period mainly covers alpha/delta circulating VoCs and only a subset of participants has been tested for BA.1 omicron sub-variant, which was instead predominant at the time of follow up. Third, these estimates strictly depend on the chosen optimal neutralizing cut-off of <1:40 which might not be meaningful for people infected with the latest Omicron sub-lineages. Lastly, comparisons of the measured levels of nAbs titers between studies are difficult because of the well-known variability in the assays used. Furthermore, although participants remain on active follow-up, we currently have no data to estimate the rate of waning of immune response past five months after the BD, and there was no monitoring of incident SARS-CoV-2 infections after vaccination by protocol. 

Key strengths of this work are the use of neutralizing activity which is currently considered the most reliable surrogate for vaccine efficacy, the large sample size and the inclusion of PLWH with a wide range of level of immunosuppression. Unfortunately, because of the efficacy of modern ART, sample size was smaller for the most important group of PLWH with CD4 count <200 cells/mm^3^.

## 5. Conclusions

Waning of humoral response against WD614G was important both four months after PVC and five months after BD although it remained above population average levels by the end of the study regardless of CD4 count. Importantly, on the contrary, humoral response against BA.1 fell on average below population average levels 5 months after BD regardless of CD4 count. The level of T cellular response was significantly higher in HCD4 and ICD4 compared to the LCD4 group although it remained above detectable levels over the entire study period regardless of CD4 count, suggesting clinical protection against severe infection, hospitalization, and death even in PLWH with immune dysfunction. Further evaluations on the effectiveness and waning of new mRNA bivalent vaccines against the current circulating omicron sub-lineages are warranted in order to establish more suitable booster vaccination strategies in case of endemic evolution of SARS-CoV-2 with increased pathogenicity.

## Figures and Tables

**Figure 1 vaccines-11-01739-f001:**
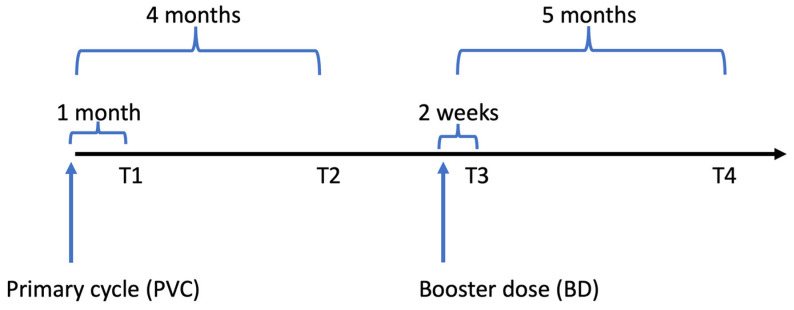
Vaccination schedule in HIV-VAC study.

**Figure 2 vaccines-11-01739-f002:**
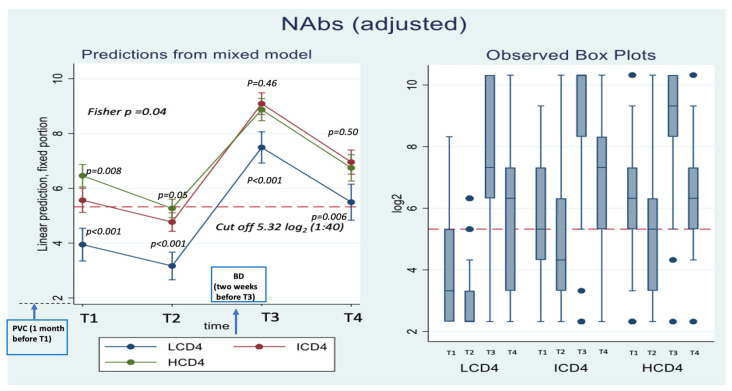
Adjusted absolute mean WD614G nAbs values over T1–T4 from fitting a mixed linear model (**left panel**) and box-plots (**right panel**) of the raw data by CD4 count groups. *Y*-axis reports nAbs values expressed in Log_2_ (cut off 5.32 log_2_); *X*-axis T1–T4 timepoints. The timepoints are: T1: 2 months after PVC, T2: 4 months after PVC, T3: 2 weeks after 3rd dose, T4: 5 months after 3rd dose.

**Figure 3 vaccines-11-01739-f003:**
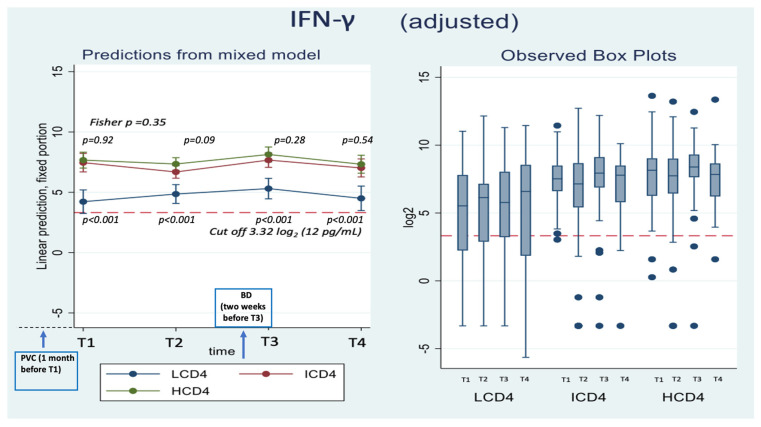
Adjusted absolute mean IFN-γ values over T1–T4 from fitting a mixed linear model (**left panel**) and boxplots (**right panel**) of the raw data by CD4 count groups. *Y*-axis reports IFN-γ values expressed in Log_2_ (cut off 3.58 log_2_); *X*-axis T1–T4 timepoints. The timepoints are: T1: 2 months after PVC, T2: 4 months after PVC, T3: 2 weeks after 3rd dose, T4: 5 months after 3rd dose.

**Table 1 vaccines-11-01739-t001:** Descriptive characteristics of the study population.

	CD4 Count at T1 ^$^ (Cells/mm^3^)				
Characteristics	LCD4 0–200/mm^3^	ICD4 201–500/mm^3^	HCD4 >500/mm^3^	*p*-Value *	Total
	N = 56	N = 120	N = 138		N = 314
** *Age, years* **				0.208	
Median (IQR)	58 (53, 63)	56 (50, 62)	56 (48, 60)		56 (50, 61)
** *Female, n (%)* **	17 (30.4)	16 (13.3)	22 (15.9)	0.018	55 (17.5)
** *Caucasian, n (%)* **	37 (66.1)	77 (64.2)	115 (83.3)	0.001	229 (72.9)
** *Nadir CD4 count, cells/mm^3^* **				<0.001	
Median (IQR)	32 (7, 70)	48 (23, 129)	166 (52, 298)		74 (26, 180)
** *Time from HIV diagnosis, years* **				0.007	
Median (IQR)	17 (2, 24)	6 (2, 13)	10 (5, 15)		9 (4, 21)
** *Time from AIDS diagnosis, years* **				0.069	
Median (IQR)	2 (1, 2)	6 (3, 7)	5 (5, 10)		5 (4, 8)
** *AIDS, n (%)* **	14 (25.9)	32 (30.2)	27 (21.3)	0.297	73 (25.4)
** *Year of starting ART* **				0.029	
Median (IQR)	2009 (2000, 2020)	2016 (2010, 2019)	2013 (2010, 2016)		2014 (2009, 2018)
** *VL≤ 50 at T1, n (%)* **	44 (78.6)	116 (97.5)	137 (99.3)	<0.001	297 (94.9)
** *Cancer, n (%)* **	2 (3.6)	6 (5.0)	12 (8.7)	0.308	20 (6.4)
** *BMI, median (IQR)* **	22 (21, 24)	24 (21, 26)	23 (22, 26)	0. 109	23 (21, 26)
** *Autoimmune disease, n (%)* **	0 (0.0)	1 (0.9)	0 (0.0)	0.426	1 (0.3)
** *Cardiopathy, n (%)* **	0 (0.0)	0 (0.0)	1 (0.8)	0.533	1 (0.3)
** *CKD, n (%)* **	8 (14.8)	9 (8.5)	9 (7.1)	0.246	26 (9.1)
** *COPD, n (%)* **	3 (5.6)	6 (5.7)	4 (3.1)	0.606	13 (4.5)
** *MI, n (%)* **	1 (1.9%)	1 (0.9%)	1 (0.8%)	0.806	3 (1.0)
** *Hypertension, n (%)* **	8 (14.8)	17 (16.0)	11 (8.7)	0.205	36 (12.5)
** *Mild liver disease, n (%)* **	12 (22.2)	23 (21.7)	29 (22.8)	0.979	64 (22.3)
** *Severe liver disease, n (%)* **	4 (7.4)	3 (2.8)	1 (0.8)	0.047	8 (2.8)
** *No. of comorbidities ^&^* **				0.189	
Median (IQR)	1 (1, 2)	1 (1, 2)	1 (1, 1)		1 (1, 2)
** *Time from T1 to T3, days* **				0.472	
Median (IQR)	178 (175, 183)	174 (162, 187)	175 (167, 199)		175 (166, 186)
** *Time from T3 to T4, days* **				0.110	
Median (IQR)	126 (122, 130)	119 (103, 133)	119 (109, 130)		120 (108, 130)

^$^ When CD4 at T1 was missing we used a last observation carried forward approach by imputing the most recent value prior to T1; T1 = time of first vaccination; ^&^ In those with ≥1 comorbidities; * Chi-square or Kruskal-Wallis test as appropriate; **Abbreviations:** LCD4, low CD4 count; ICD4, intermediate CD4 count, HCD4, high CD4 count; AIDS, Acquired Immunodeficiency syndrome; ART, antiretroviral therapy; BMI, Body Mass Index; CKD, Chronic Kidney Disease; COPD, Chronic Obstructive Pulmonary Disease; MI, Myocardial Infarction.

**Table 2 vaccines-11-01739-t002:** (**A**) Adjusted absolute mean WD614G nAbs values over T1–T4 from fitting a mixed linear model (left panel) and mean differences (right panel) by CD4 count groups. (**B**) Adjusted mean changes from T2 in WD614G nAbs values over T1–T4 from fitting a mixed linear model (left panel) and mean differences (right panel) by CD4 count groups. (**C**) Adjusted mean changes from T3 in WD614G nAbs values over T1–T4 from fitting a mixed linear model (left panel) and mean differences (right panel) by CD4 count groups.

(A)									
	Adjusted Means				Adjusted Difference in Means				
	T1 95% CI	T2 95% CI	T3 95% CI	T4 95% CI	T1 95% CI *p*-Value	T2 95% CI *p*-Value	T3 95% CI *p*-Value	T4 95% CI *p*-Value	*p*-Value *
**Nabs Whuhan**									
** *CD4 group* **									0.037
**HCD4**	6.5 (6.1, 6.9)	5.3 (4.9, 5.6)	8.8 (8.4, 9.2)	6.6 (6.1, 7.1)	0	0	0	0	
**ICD4**	5.7 (5.2, 6.1)	4.8 (4.4, 5.1)	9.1 (8.7, 9.4)	6.9 (6.4, 7.3)	−0.8 (−1.4, −0.2)	−0.5 (−1.0, −0.0)	0.2 (−0.4, 0.8)	0.2 (−0.4, 0.9)	
					0.008	0.045	0.460	0.498	
**LCD4**	3.9 (3.3, 4.5)	3.2 (2.7, 3.7)	7.5 (6.9, 8.0)	5.4 (4.8, 6.1)	−2.6 (−3.3, −1.8)	−2.1 (−2.7, −1.5)	−1.4 (−2.1, −0.7)	−1.2 (−2.0, −0.3)	
					<0.001	<0.001	<0.001	0.006	
**(B)**									
	**Adjusted ^&^ Mean Changes from T2 ^$^**				**Adjusted ^&^ Difference in Mean Changes from T2 ^$^**				
**Response**									
	**T1** **95% CI**	**T2** **95% CI**	**T3** **95% CI**	**T4** **95% CI**	**T1** **95% CI** ***p*-Value**	**T3** **95% CI** ***p*-Value**	**T4** **95% CI** ***p*-Value**	***p*-Value ***	
**Nabs Wuhan**									
** *CD4 group* **								0.011	
**HCD4**	1.2 (0.9, 1.6)	REF	3.5 (3.1, 3.8)	1.4 (1.0, 1.9)	0	0	0		
**ICD4**	1.2 (0.8, 1.6)	REF	4.3 (4.0, 4.7)	2.2 (1.9, 2.6)	−0.0 (−0.5, 0.5)	0.9 (0.4, 1.4)	0.8 (0.2, 1.4)		
					0.887	<0.001	0.006		
**LCD4**	0.9 (0.4, 1.4)	REF	4.4 (3.9, 4.8)	2.3 (1.7, 2.8)	−0.4 (−1.0, 0.2)	0.9 (0.3, 1.5)	0.8 (0.1, 1.5)		
					0.239	0.002	0.021		
**(C)**									
	**Adjusted ^&^ Mean Changes from T3** ^$$^				**Adjusted ^&^ Difference in Mean Changes from T3 ^$$^**				
**Response**									
	**T1** **95% CI**	**T2** **95% CI**	**T3** **95% CI**	**T4** **95% CI**	**T1** **95% CI** ***p*-Value**	**T2** **95% CI** ***p*-Value**	**T4** **95% CI** ***p*-Value**	***p*-Value ***	
**Nabs Wuhan**									
** *CD4 group* **								0.024	
**HCD4**	−2.6 (−3.1, −2.1)	−3.5 (−3.9, −3.1)	REF	−2.3 (−2.8, −1.8)	0	0	0		
**ICD4**	−3.6 (−4.1, −3.1)	−4.3 (−4.7, −3.9)	REF	−2.2 (−2.6, −1.8)	−1.0 (−1.7, −0.3)	−0.8 (−1.3, −0.2)	0.1 (−0.5, 0.7)		
					0.003	0.004	0.736		
**LCD4**	−3.7 (−4.2, −3.1)	−4.3 (−4.8, −3.7)	REF	−2.1 (−2.8, −1.5)	−1.1 (−1.9, −0.3)	0.2 (−0.3, 0.7)	0.2 (−0.6, 1.0)		
					0.008	0.534	0.684		

^&^ Adjusted for gender age and CD4 count nadir; * F-test type 3 interaction *p*-value; ^$^ One month after third vaccine dose; ^$$^ Two weeks after 3rd vaccine dose.

**Table 3 vaccines-11-01739-t003:** (**A**) Adjusted absolute mean IFN-γ values over T1–T4 from fitting a mixed linear model (left panel) and mean differences (right panel) by CD4 count groups. (**B**) Adjusted mean changes from T2 in IFN-γ values over T1–T4 from fitting a mixed linear model (left panel) and mean differences (right panel) by CD4 count groups.

(A)									
	Adjusted Means				Adjusted Difference in Means				
Response									
	T1 95% CI	T2 95% CI	T3 95% CI	T4 95% CI	T1 95% CI	T2 95% CI	T3 95% CI *p*-Value	T4 95% CI *p*-Value	*p*-Value *
**IFN-γ**									
** *CD4 group* **									0.34
**HCD4**	7.6 (6.9, 8.3)	7.3 (6.8, 7.9)	8.2 (7.5, 8.8)	7.5 (6.7, 8.3)	0	0	0	0	
**ICD4**	7.6 (6.8, 8.4)	6.7 (6.2, 7.2)	7.7 (7.1, 8.3)	7.1 (6.3, 7.9)	0.1 (−1.0, 1.1)	−0.7 (−1.4, 0.1)	−0.5 (−1.4, 0.4)	−0.4 (−1.5, 0.8)	
					0.917	0.089	0.280	0.543	
**LCD4**	4.2 (3.2, 5.2)	4.9 (4.1, 5.6)	5.3 (4.5, 6.2)	4.8 (3.7, 5.9)	−3.3 (−4.6, −2.1)	−2.5 (−3.5, −1.5)	−2.9 (−3.9, −1.8)	-2.7 (−4.0, −1.3)	
					<0.001	<0.001	<.001	<0.001	
**(B)**									
	**Adjusted ^&^ Mean Changes from T2 ^$^**				**Adjusted ^&^ Difference in Mean Changes from T2 ^$^**				
**Response**									
	**T1** **95% CI**	**T2** **95% CI**	**T3** **95% CI**	**T4** **95% CI**	**T1** **95% CI** ***p*-Value**	**T3** **95% CI** ***p*-Value**	**T4** **95% CI** ***p*-Value**		** *p* ** **-Value ***
**IFN-γ**									
** *CD4 group* **									0.023
**HCD4**	0.0 (−0.5, 0.6)	REF	0.8 (0.3, 1.3)	−0.2 (−0.8, 0.4)	0	0	0		
**ICD4**	1.2 (0.5, 1.8)	REF	1.0 (0.6, 1.5)	0.3 (−0.3, 0.9)	1.1 (0.3, 2.0)	0.2 (−0.5, 0.9)	0.5 (−0.4, 1.4)		
					0.011	0.559	0.271		
**LCD4**	−1.0 (−1.8, −0.2)	REF	0.5 (−0.1, 1.2)	−0.7 (−1.5, 0.2)	−1.0 (−2.0, −0.0)	−0.3 (−1.2, 0.5)	−0.5 (−1.5, 0.6)		
					0.040	0.450	0.376		

^&^ Adjusted for gender age and CD4 count nadir; * F-test type 3 interaction *p*-value; ^$^ One month after 3rd vaccine dose.

## Data Availability

Data used in this analysis is available upon reasonable request.

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
