# Peer review of "Long Term Assessment of Anti-SARS-CoV-2 Immunogenicity after mRNA Vaccine in Persons Living with HIV"

_vaccines, 2023, doi:10.3390/vaccines11121739_

Round 1
Reviewer 1 Report
Comments and Suggestions for Authors
Vergori et al. report on the durability of vaccine-induced immune responses in people living with HIV with distinct CD4 T cell counts. The manuscript addresses a novel question. However, the submission would benefit from edits and clarifications.
Major
1 - The results section reveals that some participants received an initial adenoviral vector vaccine. This information is currently absent from the methods section.
2 - No evidence demonstrates that the method used to measure T-cell IFN-gamma production detects only T-cell responses. Indeed, a similar method is used in the HIV literature to detect anti-HIV ADCC responses mediated by NK cells. The addition of peptides to whole blood containing anti-viral antibodies can trigger NK cell IFN-gamma production (see Parsons et al. 2012 - PMID: 22345455 ). If the authors are to claim that this method detects T-cell responses, an additional experiment is needed to confirm that IFN-gamma is not being produced by cell types other than T-cells.
3 - Were CD4 counts measured throughout the study? Can the authors comment on how stable the counts were and if any changes throughout the study could impact data interpretation?
4 - The figures should be amended to highlight differences between the groups in the assessed variables at distinct time points.
Minor
1 - For the manuscript title, PLWH should be spelled out (i.e., people living with HIV). A more general scientific audience does not know this acronym.
2 - The manuscript needs editing to shorten run-on sentences (e.g., the first two paragraphs of the paper) and fix various typos (e.g., "T-cells immunity" on line 58, "was approved" on line 87, "P-value" line 226, etc).
3 - P-value is missing on line 223 - "p=???".
Comments on the Quality of English LanguageIn general, the manuscript is easy to understand. However, various long sentences should be revised. For example, the first paragraph of the paper is one sentence. It would benefit the reader for such sentences to be broken into smaller sentences.
The manuscript also has various typos that should be corrected.
Author Response
Vergori et al. report on the durability of vaccine-induced immune responses in people living with HIV with distinct CD4 T cell counts. The manuscript addresses a novel question. However, the submission would benefit from edits and clarifications.
Major points
1 – The results section reveals that some participants received an initial adenoviral vector vaccine. This information is currently absent from the methods section.
We thank the reviewer for the comment. Indeed, 1% of the study population (n=5) was vaccinated with an adenoviral vaccine. We have now added the information in the Methods section (lines 101-102).
2 - No evidence demonstrates that the method used to measure T-cell IFN-gamma production detects only T-cell responses. Indeed, a similar method is used in the HIV literature to detect anti-HIV ADCC responses mediated by NK cells. The addition of peptides to whole blood containing anti-viral antibodies can trigger NK cell IFN-gamma production (see Parsons et al. 2012 - PMID: 22345455 ). If the authors are to claim that this method detects T-cell responses, an additional experiment is needed to confirm that IFN-gamma is not being produced by cell types other than T-cells.
We thank the reviewer fort the comment. The IGRA assay has been used to monitor the antigen specific T cell response in many papers regarding SARS-CoV2 infection and/or vaccination (Vergori et al 2022- doi: 10.1038/s41467-022-32263-7; Agrati at al 2021-https://www.mdpi.com/2076-2607/9/6/1315; Corradini et al 2023- doi: 10.1093/cid/ciac404; Tortorella et al 2022-doi: 10.1212/WNL.0000000000013108; D’Offizi et al 2022-doi: 10.1111/liv.15089. Epub 2021 Nov 17; Saraceni et al 2022-doi: 10.3390/diseases10030049; Aiello et al 2022-doi: 10.1016/j.ijid.2022.07.049; Strengert et al 2021-doi: 10.1016/j.ebiom.2021.103524; Brisotto G et al, 2023-doi: 10.3390/v15061276; Santoro et al, 2023 doi: 10.3390/biomedicines11051247; Aiello et al, 2023-DOI: 10.1136/jnnp-2022-330175) as well as in other microbial infection (e.g. mycobacterial Tuberculosis). During the setting up of this procedure, we checked its ability to specifically stimulate CD3 T cells. After the reviewer’s request, we repeated the flow cytometric assay in two healthy donors to verify the phenotype of IFN-g-producing cells after peptides stimulation. As you can see in the Figure below, we observed that most of the IFN-gamma are produced by CD3+ T cells. We expect similar results in PLWH.
3 – Were CD4 counts measured throughout the study? Can the authors comment on how stable the counts were and if any changes throughout the study could impact data interpretation?
For 238 participants the CD4 count at T1 was used. For the remaining participants this value was missing and therefore has been replaced by the value at previous dose vaccination (n=34, 10%) and by the value prior to the start of the vaccination campaign (n=42, 13%). To control for CD4 fluctuation before any of these time points, all models have been controlled for participants’ CD4 count nadir. We have now also repeated the main analysis after restricting it to the 238 participants for which there were no missing CD4 count data at T1, and results were very consistent with those of the main analysis as shown below.
Table 2A – revised after restricting to 238 participants with complete CD4 count data.
|
|
Visits |
|
|||||||
|
|
Unadjusted means |
Unadjusted difference in means |
|
||||||
|
Response |
|
|
|
|
|
|
|
|
|
|
|
T1 |
T2 |
T3 |
T4 |
T1 |
T2 |
T3 |
T4 |
p-value* |
|
NAbs Whuhan |
|
|
|
|
|
|
|
|
|
|
CD4 group |
|
|
|
|
|
|
|
|
0.032 |
|
HCD4 |
6.5 (6.1, 6.9) |
5.2 (4.9, 5.6) |
8.8 (8.4, 9.2) |
6.5 (6.0, 7.0) |
0 |
0 |
0 |
0 |
|
|
ICD4 |
5.7 (5.3, 6.2) |
4.8 (4.4, 5.2) |
9.2 (8.8, 9.7) |
7.0 (6.5, 7.4) |
-0.7 (-1.4, -0.1) |
-0.4 (-1.0, 0.1) |
0.4 (-0.2, 1.0) |
0.4 (-0.3, 1.2) |
|
|
|
|
|
|
|
0.017 |
0.096 |
0.166 |
0.223 |
|
|
LCD4 |
4.0 (3.4, 4.6) |
3.2 (2.6, 3.9) |
7.4 (6.8, 8.1) |
5.4 (4.7, 6.1) |
-2.5 (-3.2, -1.7) |
-2.0 (-2.7, -1.3) |
-1.4 (-2.1, -0.6) |
-1.2 (-2.0, -0.3) |
|
|
|
|
|
|
|
<.001 |
<.001 |
<.001 |
0.010 |
|
|
*F-test type 3 interaction p-value |
|||||||||
Table 3A – revised after restricting to 238 participants with complete CD4 count data.
|
|
Visits |
|
||||||
|
|
Unadjusted mean changes from T2$ |
Unadjusted difference in mean changes from T2$ |
|
|||||
|
Response |
|
|
|
|
|
|
|
|
|
|
T1 |
T2 |
T3 |
T4 |
T1 |
T3 |
T4 |
p-value* |
|
NAbs Wuhan |
|
|
|
|
|
|
|
|
|
CD4 group |
|
|
|
|
|
|
|
0.009 |
|
HCD4 |
1.3 (1.0, 1.7) |
-0.0 (-0.3, 0.3) |
3.5 (3.1, 3.9) |
1.3 (0.9, 1.8) |
0 |
0 |
0 |
|
|
ICD4 |
1.2 (0.8, 1.6) |
0.0 (-0.3, 0.3) |
4.4 (4.1, 4.8) |
2.1 (1.7, 2.5) |
-0.2 (-0.7, 0.4) |
0.9 (0.4, 1.5) |
0.8 (0.2, 1.4) |
|
|
|
|
|
|
|
0.569 |
<.001 |
0.013 |
|
|
LCD4 |
0.8 (0.3, 1.3) |
-0.0 (-0.5, 0.5) |
4.2 (3.7, 4.8) |
2.2 (1.6, 2.7) |
-0.5 (-1.1, 0.1) |
0.7 (0.1, 1.4) |
0.8 (0.1, 1.6) |
|
|
|
|
|
|
|
0.100 |
0.028 |
0.030 |
|
|
$Pre 3rd dose |
||||||||
|
*F-test type 3 interaction p-value |
||||||||
4 - The figures should be amended to highlight differences between the groups in the assessed variables at distinct time points.
The p-values were already shown in the Tables 2A and 3A and to avoid redundancy had not been added in the Figures. We have now added theme in the Figures as requested.
Minor points
1 - For the manuscript title, PLWH should be spelled out (i.e., people living with HIV). A more general scientific audience does not know this acronym.
We thank the reviewer for this suggestion. The title has been modified accordingly.
2 - The manuscript needs editing to shorten run-on sentences (e.g., the first two paragraphs of the paper) and fix various typos (e.g., "T-cells immunity" on line 58, "was approved" on line 87, "P-value" line 226, etc).
We thank the reviewer for the notification. We tried to correct all typos and sent the manuscript to an English editing service.
3 - P-value is missing on line 223 - "p=???".
We thank the reviewer for spotting this, it was an oversight. We have now inserted the missing p-values as shown below:
Values of spike-specific T cell response in HCD4 and ICD4 were higher than those seen for LCD4 (for example at T3 : 8.2 log2 HCD4 vs 7.7 (p=0.09) ICD4 vs 5.3 LCD4 p<0.001; Table 3A, Figure 3)[…].
Comments on the Quality of English Language
In general, the manuscript is easy to understand. However, various long sentences should be revised. For example, the first paragraph of the paper is one sentence. It would benefit the reader for such sentences to be broken into smaller sentences.
The manuscript also has various typos that should be corrected.
We agree with this reviewer, and we have tried to shorten the long sentences and improve the paper readability in general. We hope that we have also corrected all the typos.
Reviewer 2 Report
Comments and Suggestions for Authors
Vergori et al. propose a biological study assessing humoral and cellular responses to the mRNA COVID-19 vaccine at 4 time points post vaccination and boost in people living with HIV stratified according to CD4 cell counts. They found that the nAb to WD614G/Omicron BA.1 SARS-CoV-2 wanes more rapidly in the Low CD4 group, and that the level of cellular responses to SARS-CoV-2 spike is lower for the Low CD4 group than in the Intermediate and High CD4 groups.
The study is quite well designed and well presented but several points need to be clarified before publication:
Major points
- It would have be very informative to have a HIV-negative control group to evaluate if the HCD4 group behaves as HIV-negative counterparts. Maybe the authors could at least quote the results with the same techniques for healthy donors.
- The number of case in the LCD4 group is low when compared to ICD4 and HCD4 groups.
- The design is not clear enough: were there several clinical centers in Italy, or were all the patients included in the same center?
- Did the LCD4 group included in hospitalised patients? Did they have opportunistic infections at the time of vaccination?
- Concerning the IFNgamma assay : what was the blood volume used for stimulation? Is the test as reproductible as a Quantiferon assay? Were quality controls tested at the beginning and end of the experiments? The test was not adjusted for lymphocytes counts so the IFNgamma levels could have been underestimated in the LCD4 group.
- The inclusions were not prospective but patients were included after vaccination.
- The number of COVID-19 cases after vaccination should be quoted.
- Is there any relationship between the nAb response and the IFNg response? The correlation could be studied in order to identify good responders and their CD4 group belonging. Are the bad responders those with detectable HIV-RNA levels?
Minor points
- The third sentence of the conclusion in the abstract should rather fit in the results paragraph.
- The methods for the CD4 cell counts should be described.
- Page 5, line 223, p=???: what is the p value?
- Table 1: P values are not in line with other columns.
- Table 2A: values in comas are IQR or min-max?
- Figure 1: depict the vaccination on the figure.
Comments on the Quality of English LanguageParagraphs need to be reorganized.
Author Response
Vergori et al. propose a biological study assessing humoral and cellular responses to the mRNA COVID-19 vaccine at 4 time points post vaccination and boost in people living with HIV stratified according to CD4 cell counts. They found that the nAb to WD614G/Omicron BA.1 SARS-CoV-2 wanes more rapidly in the Low CD4 group, and that the level of cellular responses to SARS-CoV-2 spike is lower for the Low CD4 group than in the Intermediate and High CD4 groups.
The study is quite well designed and well-presented, but several points need to be clarified before publication:
Major points
- It would have be very informative to have a HIV-negative control group to evaluate if the HCD4 group behaves as HIV-negative counterparts. Maybe the authors could at least quote the results with the same techniques for healthy donors.
Unfortunately, we did not have the opportunity of including a control group in this analysis. However, the same group of virologists and immunologists performed a similar study assessing the humoral and T cellular response in a cohort of healthy individuals without HIV infection (health care workers -HCW) who were vaccinated with anti-SARS-CoV-2 mRNA vaccines. In this parallel study a significant reduction of anti-RBD titre was observed after 3 months and a waning kinetic in neutralizing antibody which was similar to that seen in our HCD4 group. Specifically, all HCWs showed a positive neutralization test 3 months after vaccination, but the titre started to wane after this point in time. Also, in this cohort of HCW, similarly to what seen in our HIV-infected participants, specific T cells responses to mRNA vaccines have instead persisted overtime [17]. Moreover, we also compared post 3rd dose titres in the HCD4 group with those measured in a cohort of HCW within a study conducted at our Institute using the same methods [38] and again data carried no evidence for a difference: D614G nAbs titres in HCD4 2-weeks after the 3rd dose were 8.8 log2(T3) versus 8.3 log2 (1:320) in HCWs at 1 month post 3rd dose. Also, no evidence for a difference was detected for IFN-gamma: 8.2 log2 in HCD4 vs 8.3 log2 in HCWs.
We have added a sentence in the Discussion section to link our results with these data.
- The number of cases in the LCD4 group is low when compared to ICD4 and HCD4 groups.
Unfortunately, we could not enrol many participants in the LCD4 as opposed to the other groups. These is because nowadays most PLWH respond well to ART and their current CD4 count is typically >200 cells/mm3 Despite this limitation the analysis (which has been now acknowledged in the Discussion section) was adequately powered to detect many differences between the LCD4 and the HCD4 group.
The design is not clear enough: were there several clinical centres in Italy, or were all the patients included in the same centre?
We apologise if the study design had not clearly presented. It is an observational, monocentric study, enrolments and all tests have been performed at a single Institute (the Lazzaro Spallanzani Hospital in Rome). We have now tried to better clarify this in the Method section.
- Did the LCD4 group included in hospitalised patients? Did they have opportunistic infections at the time of vaccination?
None of the included participants were currently hospitalised and if they previous had opportunistic infections they could not be enrolled in the vaccination protocol unless they have been successfully treated and had recovered from the illness. We have clarified this in the Methods.
- Concerning the IFNgamma assay : what was the blood volume used for stimulation? Is the test as reproductible as a Quantiferon assay? Were quality controls tested at the beginning and end of the experiments? The test was not adjusted for lymphocytes counts so the IFNgamma levels could have been underestimated in the LCD4 group.
We used 1 mL of whole blood for each setting (unstimulated, Spike and SEB). Moreover, we verified the reproducibility of this test by analysing the coefficient of variation of 8 replicates of two different healthy controls: a) HD1 coefficient of variation: 33.1/148.2=0.22, 22%; b) HD2 coefficient of variation: 12.9/62.0=0.20, 20%.
Several control steps in our experiments have been optimized to ensure high quality of results:
- i) the whole blood was plated within 3 hours from the blood draw, immediately stimulated with the peptides and SEB and transferred in the incubator.
- ii) the vials containing the spike peptides and SEB were stored at -80°C until use
iii) a control plasma containing a known amount of IFN-g is added in each ELISA experiments
We agree with the referees that the results were not normalized for the number of lymphocytes. Nevertheless, in our experience, the direct stimulation of whole blood represents an optimal approach to mirror the “real” immunity in each patient.
Indeed, the patients with a high lymphocyte count show a higher probability to have a good spike specific T cell response than patients with a low lymphocytes count. But this is what happens in vivo. If we normalized the IFN-g values on the number of lymphocytes, we would reduce the differences that exist in vivo.
- The inclusions were not prospective, but patients were included after vaccination.
We are not sure we understood this comment. The study is a prospective cohort of PLWH who were consecutively enrolled into a vaccination programme and followed up through a vaccination schedule requiring sample storage after each dose of the vaccine and retrospective analysis of the sample to measure vaccine response.
- The number of COVID-19 cases after vaccination should be quoted.
Participants who had evidence of being infected with SARS CoV-2 prior to the start of the vaccination campaign have been excluded. Unfortunately, although a minority who happened to have evidence of infection during the vaccination program were also excluded, our protocol was not designed to monitor for infections over follow-up. We have added this limitation in the Discussion section.
- Is there any relationship between the nAb response and the IFNg response? The correlation could be studied in order to identify good responders and their CD4 group belonging. Are the bad responders those with detectable HIV-RNA levels?
We have now evaluated the relationship between nAbs and IFNg response both at time T1 and T3, overall and after stratification for CD4 count groups. The results of these additional analyses are shown below and are also now included as Supplementary material.
The Table shows that there was a significant correlation between responses (p<0.0001), especially in the cross-sectional analyses correlating values measured at the same time and was stronger at T1 (rho=0.48) vs. T3 (rho=0.37). The plot, after stratification for CD4 count groups, and consistently with the main results shows that lower responses in both nAbs and IFN-gamma were seen in the LCD4 group vs. the HCD4 group, also leading to stronger correlations with steeper slopes. More details about this analysis could be show upon request.
Table. Matrix of Spearman rho correlation coefficients
|
|
nAbs at T1 |
nAbs at T3 |
IFN-γ at T1 |
|
nAbs at T1 |
|
|
|
|
nAbs at T3 |
0.72 P<0.0001 |
|
|
|
IFN-γ at T1 |
0.48 P<0.0001 |
0.37 P<0.0001 |
|
|
IFN-γ at T3 |
0.24 P=0.002 |
0.29 P<0.0001 |
0.68 P<0.0001 |
Figure 1. Correlation at time T1
Figure 2. Correlation at time T3
Minor points
- The third sentence of the conclusion in the abstract should rather fit in the results paragraph.
We have reworded the sentence so that it is no longer a mere repetition of the sentence in the Results.
- The methods for the CD4 cell counts should be described.
CD4-T cell count was meaured by flow cytometry (Aquios, Beckman Coulter, CA, USA) [24]. This sentence has been added in method section.
- Page 5, line 223, p=???: what is the p value?
Thank you for spotting this, it was an oversight. We have now inserted the missing p-values as shown below:
Values of spike-specific T cell response in HCD4 and ICD4 were higher than those seen for LCD4 (for example at T3 : 8.2 log2 HCD4 vs 7.7 (p=0.09) ICD4 vs 5.3 LCD4 p<0.001; Table 3A, Figure 3) […].
- Table 1: P values are not in line with other columns.
We checked the alignment, and all seem to be correct. Of note, for continuous variables or categorical variables with more than two groups, the p-value has been deliberately situated on the same line as the variable name instead of against the median or percentages values.
- Table 2A: values in comas are IQR or min-max?
No, these are the 95% CI from fitting the mixed linear model as indicated in the header.
- Figure 1: depict the vaccination on the figure.
We have now added an additional Figure which depicts the vaccination and sampling storage schedule after each dose vaccination (Figure 1)
Comments on the Quality of English Language.Paragraphs need to be reorganized.
We have shortened the long sentence and reorganised some of the paragraphs in the Introduction and Discussion sections to improve the paper readability as suggested.
Round 2
Reviewer 1 Report
Comments and Suggestions for Authors
The authors have addressed my concerns.
Author Response
We thank the reviewer for his/her valuable review.
Reviewer 2 Report
Comments and Suggestions for Authors
Vergori et al. propose a biological study assessing humoral and cellular responses to the mRNA COVID-19 vaccine at 4 time points post vaccination and boost in people living with HIV stratified according to CD4 cell counts. They found that the nAb to WD614G/Omicron BA.1 SARS-CoV-2 wanes more rapidly in the Low CD4 group, and that the level of cellular responses to SARS-CoV-2 spike is lower for the Low CD4 group than in the Intermediate and High CD4 groups.
The study is quite well designed and well-presented, but several points need to be clarified before publication:
Major points
- It would have be very informative to have a HIV-negative control group to evaluate if the HCD4 group behaves as HIV-negative counterparts. Maybe the authors could at least quote the results with the same techniques for healthy donors.
Unfortunately, we did not have the opportunity of including a control group in this analysis. However, the same group of virologists and immunologists performed a similar study assessing the humoral and T cellular response in a cohort of healthy individuals without HIV infection (health care workers -HCW) who were vaccinated with anti-SARS-CoV-2 mRNA vaccines.
è Are the techniques used the same in HCW and PLWH? Please clarifiy in the discussion.
In this parallel study a significant reduction of anti-RBD titre was observed after 3 months and a waning kinetic in neutralizing antibody which was similar to that seen in our HCD4 group. Specifically, all HCWs showed a positive neutralization test 3 months after vaccination, but the titre started to wane after this point in time. Also, in this cohort of HCW, similarly to what seen in our HIV-infected participants, specific T cells responses to mRNA vaccines have instead persisted overtime [17]. Moreover, we also compared post 3rd dose titres in the HCD4 group with those measured in a cohort of HCW within a study conducted at our Institute using the same methods [38] and again data carried no evidence for a difference: D614G nAbs titres in HCD4 2-weeks after the 3rd dose were 8.8 log2(T3) versus 8.3 log2 (1:320) in HCWs at 1 month post 3rd dose.
Also, no evidence for a difference was detected for IFN-gamma: 8.2 log2 in HCD4 vs 8.3 log2 in HCWs.
è Please add this information in the discussion.
We have added a sentence in the Discussion section to link our results with these data.
- The number of cases in the LCD4 group is low when compared to ICD4 and HCD4 groups.
Unfortunately, we could not enrol many participants in the LCD4 as opposed to the other groups. These is because nowadays most PLWH respond well to ART and their current CD4 count is typically >200 cells/mm3 Despite this limitation the analysis (which has been now acknowledged in the Discussion section) was adequately powered to detect many differences between the LCD4 and the HCD4 group.
The design is not clear enough: were there several clinical centres in Italy, or were all the patients included in the same centre?
We apologise if the study design had not clearly presented. It is an observational, monocentric study, enrolments and all tests have been performed at a single Institute (the Lazzaro Spallanzani Hospital in Rome). We have now tried to better clarify this in the Method section.
- Did the LCD4 group included in hospitalised patients? Did they have opportunistic infections at the time of vaccination?
None of the included participants were currently hospitalised and if they previous had opportunistic infections they could not be enrolled in the vaccination protocol unless they have been successfully treated and had recovered from the illness. We have clarified this in the Methods.
- Concerning the IFNgamma assay : what was the blood volume used for stimulation? Is the test as reproductible as a Quantiferon assay? Were quality controls tested at the beginning and end of the experiments? The test was not adjusted for lymphocytes counts so the IFNgamma levels could have been underestimated in the LCD4 group.
We used 1 mL of whole blood for each setting (unstimulated, Spike and SEB). Moreover, we verified the reproducibility of this test by analysing the coefficient of variation of 8 replicates of two different healthy controls: a) HD1 coefficient of variation: 33.1/148.2=0.22, 22%; b) HD2 coefficient of variation: 12.9/62.0=0.20, 20%.
è What are the differences with Quantiferon settings, what is the Spike peptides concentration for whole blood stimulation?
Several control steps in our experiments have been optimized to ensure high quality of results:
i) the whole blood was plated within 3 hours from the blood draw, immediately stimulated with the peptides and SEB and transferred in the incubator.
ii) the vials containing the spike peptides and SEB were stored at -80°C until use
iii) a control plasma containing a known amount of IFN-g is added in each ELISA experiments
We agree with the referees that the results were not normalized for the number of lymphocytes. Nevertheless, in our experience, the direct stimulation of whole blood represents an optimal approach to mirror the “real” immunity in each patient.
Indeed, the patients with a high lymphocyte count show a higher probability to have a good spike specific T cell response than patients with a low lymphocytes count. But this is what happens in vivo. If we normalized the IFN-g values on the number of lymphocytes, we would reduce the differences that exist in vivo.
- The inclusions were not prospective, but patients were included after vaccination.
We are not sure we understood this comment. The study is a prospective cohort of PLWH who were consecutively enrolled into a vaccination programme and followed up through a vaccination schedule requiring sample storage after each dose of the vaccine and retrospective analysis of the sample to measure vaccine response.
- The number of COVID-19 cases after vaccination should be quoted.
Participants who had evidence of being infected with SARS CoV-2 prior to the start of the vaccination campaign have been excluded. Unfortunately, although a minority who happened to have evidence of infection during the vaccination program were also excluded, our protocol was not designed to monitor for infections over follow-up. We have added this limitation in the Discussion section.
- Is there any relationship between the nAb response and the IFNg response? The correlation could be studied in order to identify good responders and their CD4 group belonging. Are the bad responders those with detectable HIV-RNA levels?
We have now evaluated the relationship between nAbs and IFNg response both at time T1 and T3, overall and after stratification for CD4 count groups. The results of these additional analyses are shown below and are also now included as Supplementary material.
The Table shows that there was a significant correlation between responses (p<0.0001), especially in the cross-sectional analyses correlating values measured at the same time and was stronger at T1 (rho=0.48) vs. T3 (rho=0.37). The plot, after stratification for CD4 count groups, and consistently with the main results shows that lower responses in both nAbs and IFN-gamma were seen in the LCD4 group vs. the HCD4 group, also leading to stronger correlations with steeper slopes. More details about this analysis could be show upon request.
è New Figure 1 is not clear enough, rather indicate the vaccinations on previous Figure 1 and Figure 2.
Comments on the Quality of English LanguageEnglish langage seems correct.
Author Response
Vergori et al. propose a biological study assessing humoral and cellular responses to the mRNA COVID-19 vaccine at 4 time points post vaccination and boost in people living with HIV stratified according to CD4 cell counts. They found that the nAb to WD614G/Omicron BA.1 SARS-CoV-2 wanes more rapidly in the Low CD4 group, and that the level of cellular responses to SARS-CoV-2 spike is lower for the Low CD4 group than in the Intermediate and High CD4 groups.
The study is quite well designed and well-presented, but several points need to be clarified before publication:
Major points
- It would have be very informative to have a HIV-negative control group to evaluate if the HCD4 group behaves as HIV-negative counterparts. Maybe the authors could at least quote the results with the same techniques for healthy donors.
Unfortunately, we did not have the opportunity of including a control group in this analysis. However, the same group of virologists and immunologists performed a similar study assessing the humoral and T cellular response in a cohort of healthy individuals without HIV infection (health care workers -HCW) who were vaccinated with anti-SARS-CoV-2 mRNA vaccines.
è Are the techniques used the same in HCW and PLWH? Please clarifiy in the discussion.
Yes, they are. Methods are clarified already in the discussion section in the previously submitted revised version (see line 335) and added on line 371.
In this parallel study a significant reduction of anti-RBD titre was observed after 3 months and a waning kinetic in neutralizing antibody which was similar to that seen in our HCD4 group. Specifically, all HCWs showed a positive neutralization test 3 months after vaccination, but the titre started to wane after this point in time. Also, in this cohort of HCW, similarly to what seen in our HIV-infected participants, specific T cells responses to mRNA vaccines have instead persisted overtime [17]. Moreover, we also compared post 3rd dose titres in the HCD4 group with those measured in a cohort of HCW within a study conducted at our Institute using the same methods [38] and again data carried no evidence for a difference: D614G nAbs titres in HCD4 2-weeks after the 3rd dose were 8.8 log2(T3) versus 8.3 log2 (1:320) in HCWs at 1 month post 3rd dose.
Also, no evidence for a difference was detected for IFN-gamma: 8.2 log2 in HCD4 vs 8.3 log2 in HCWs.
è Please add this information in the discussion.
Thank you, it has been added in discussion section (see lines 373-374).
We have added a sentence in the Discussion section to link our results with these data.
- The number of cases in the LCD4 group is low when compared to ICD4 and HCD4 groups.
Unfortunately, we could not enrol many participants in the LCD4 as opposed to the other groups. These is because nowadays most PLWH respond well to ART and their current CD4 count is typically >200 cells/mm3 Despite this limitation the analysis (which has been now acknowledged in the Discussion section) was adequately powered to detect many differences between the LCD4 and the HCD4 group.
The design is not clear enough: were there several clinical centres in Italy, or were all the patients included in the same centre?
We apologise if the study design had not clearly presented. It is an observational, monocentric study, enrolments and all tests have been performed at a single Institute (the Lazzaro Spallanzani Hospital in Rome). We have now tried to better clarify this in the Method section.
- Did the LCD4 group included in hospitalised patients? Did they have opportunistic infections at the time of vaccination?
None of the included participants were currently hospitalised and if they previous had opportunistic infections they could not be enrolled in the vaccination protocol unless they have been successfully treated and had recovered from the illness. We have clarified this in the Methods.
- Concerning the IFNgamma assay : what was the blood volume used for stimulation? Is the test as reproductible as a Quantiferon assay? Were quality controls tested at the beginning and end of the experiments? The test was not adjusted for lymphocytes counts so the IFNgamma levels could have been underestimated in the LCD4 group.
We used 1 mL of whole blood for each setting (unstimulated, Spike and SEB). Moreover, we verified the reproducibility of this test by analysing the coefficient of variation of 8 replicates of two different healthy controls: a) HD1 coefficient of variation: 33.1/148.2=0.22, 22%; b) HD2 coefficient of variation: 12.9/62.0=0.20, 20%.
è What are the differences with Quantiferon settings, what is the Spike peptides concentration for whole blood stimulation?
The difference with QuantiFERON is that the blood is stimulated in plate and not in tube and that the peptides are from SARS-CoV-2 and not MTB. Final peptide concentration 0.1ug/ml
Several control steps in our experiments have been optimized to ensure high quality of results:
- i) the whole blood was plated within 3 hours from the blood draw, immediately stimulated with the peptides and SEB and transferred in the incubator.
- ii) the vials containing the spike peptides and SEB were stored at -80°C until use
iii) a control plasma containing a known amount of IFN-g is added in each ELISA experiments
We agree with the referees that the results were not normalized for the number of lymphocytes. Nevertheless, in our experience, the direct stimulation of whole blood represents an optimal approach to mirror the “real” immunity in each patient.
Indeed, the patients with a high lymphocyte count show a higher probability to have a good spike specific T cell response than patients with a low lymphocytes count. But this is what happens in vivo. If we normalized the IFN-g values on the number of lymphocytes, we would reduce the differences that exist in vivo.
- The inclusions were not prospective, but patients were included after vaccination.
We are not sure we understood this comment. The study is a prospective cohort of PLWH who were consecutively enrolled into a vaccination programme and followed up through a vaccination schedule requiring sample storage after each dose of the vaccine and retrospective analysis of the sample to measure vaccine response.
- The number of COVID-19 cases after vaccination should be quoted.
Participants who had evidence of being infected with SARS CoV-2 prior to the start of the vaccination campaign have been excluded. Unfortunately, although a minority who happened to have evidence of infection during the vaccination program were also excluded, our protocol was not designed to monitor for infections over follow-up. We have added this limitation in the Discussion section.
- Is there any relationship between the nAb response and the IFNg response? The correlation could be studied in order to identify good responders and their CD4 group belonging. Are the bad responders those with detectable HIV-RNA levels?
We have now evaluated the relationship between nAbs and IFNg response both at time T1 and T3, overall and after stratification for CD4 count groups. The results of these additional analyses are shown below and are also now included as Supplementary material.
The Table shows that there was a significant correlation between responses (p<0.0001), especially in the cross-sectional analyses correlating values measured at the same time and was stronger at T1 (rho=0.48) vs. T3 (rho=0.37). The plot, after stratification for CD4 count groups, and consistently with the main results shows that lower responses in both nAbs and IFN-gamma were seen in the LCD4 group vs. the HCD4 group, also leading to stronger correlations with steeper slopes. More details about this analysis could be show upon request.
è New Figure 1 is not clear enough, rather indicate the vaccinations on previous Figure 1 and Figure 2.
We have fixed figure 1, there was a misalignment by mistake. We also added vaccination time on the other 2 figures.
Comments on the Quality of English Language
English langage seems correct